# Triterpene Glycosides from the Far Eastern Sea Cucumber *Psolus chitonoides*: Chemical Structures and Cytotoxicities of Chitonoidosides E_1_, F, G, and H

**DOI:** 10.3390/md19120696

**Published:** 2021-12-07

**Authors:** Alexandra S. Silchenko, Anatoly I. Kalinovsky, Sergey A. Avilov, Pelageya V. Andrijaschenko, Roman S. Popov, Ekaterina A. Chingizova, Vladimir I. Kalinin, Pavel S. Dmitrenok

**Affiliations:** G.B. Elyakov Pacific Institute of Bioorganic Chemistry, Far Eastern Branch of the Russian Academy of Sciences, Pr. 100-letya Vladivostoka 159, 690022 Vladivostok, Russia; silchenko_als@piboc.dvo.ru (A.S.S.); kaaniv@piboc.dvo.ru (A.I.K.); avilov_sa@piboc.dvo.ru (S.A.A.); andrijashchenko_pv@piboc.dvo.ru (P.V.A.); popov_rs@piboc.dvo.ru (R.S.P.); chingizova_ea@piboc.dvo.ru (E.A.C.)

**Keywords:** *Psolus chitonoides*, triterpene glycosides, chitonoidosides, sea cucumber, cytotoxic activity

## Abstract

Four new triterpene disulfated glycosides, chitonoidosides E_1_ (**1**), F (**2**), G (**3**), and H (**4**), were isolated from the Far-Eastern sea cucumber *Psolus chitonoides* and collected near Bering Island (Commander Islands) at depths of 100–150 m. Among them there are two hexaosides (**1** and **3**), differing from each other by the terminal (sixth) sugar residue, one pentaoside (**4**) and one tetraoside (**2**), characterized by a glycoside architecture of oligosaccharide chains with shortened bottom semi-chains, which is uncommon for sea cucumbers. Some additional distinctive structural features inherent in **1**–**4** were also found: the aglycone of a recently discovered new type, with 18(20)-ether bond and lacking a lactone in chitonoidoside G (**3**), glycoside 3-*O*-methylxylose residue in chitonoidoside E_1_ (**1**), which is rarely detected in sea cucumbers, and sulfated by uncommon position 4 terminal 3-*O*-methylglucose in chitonoidosides F (**2**) and H (**4**). The hemolytic activities of compounds **1**–**4** and chitonoidoside E against human erythrocytes and their cytotoxic action against the human cancer cell lines, adenocarcinoma HeLa, colorectal adenocarcinoma DLD-1, and monocytes THP-1, were studied. The glycoside with hexasaccharide chains (**1**, **3** and chitonoidoside E) were the most active against erythrocytes. A similar tendency was observed for the cytotoxicity against adenocarcinoma HeLa cells, but the demonstrated effects were moderate. The monocyte THP-1 cell line and erythrocytes were comparably sensitive to the action of the glycosides, but the activity of chitonoidosides E and E_1_ (**1**) significantly differed from that of **3** in relation to THP-1 cells. A tetraoside with a shortened bottom semi-chain, chitonoidoside F (**2**), displayed the weakest membranolytic effect in the series.

## 1. Introduction

Triterpene glycosides are characteristic secondary metabolites of the sea cucumbers. Extensive studies on glycosides provide significant information on the exploration of chemical diversity, properties and biological activity of a huge collection of natural products, which are a valuable and promising resource of new drugs and medicines [1,2,3,4,5,6,7,8]. The interest in these compounds is also driven by their taxonomic specificity [9,10,11] as well as the possibility of reconstructing the sequences of the biosynthetic transformations of aglycones and carbohydrate chains during biosynthesis [12,13] and of defining the peculiarities of «structure-activity relationships» based on knowledge about their structural diversity [14]. All this indicates the relevance of searching for new glycosides. The Far Eastern sea cucumber *Psolus chitonoides* is the fourth chemically studied representative of the genus *Psolus*. The animals of this species contain a complicated multicomponent mixture of triterpene glycosides. Therefore, their separation and purification are difficult and time-consuming. Recently, we published a paper concerning the isolation, structural elucidation, and biologic activity of a series of the glycosides, named chitonoidosides A–E, isolated from *P. chitonoides* [15]. These compounds feature some interesting structural features, such as a new, non-holostane aglycone lacking a lactone and featuring an 18(20)-epoxy cycle, 3-*O*-methylxylose residue in the carbohydrate chains of three of them, the sulfation of 3-*O*-methylxylose by C-4, and, finally, a rather rare architecture of tetrasaccharide carbohydrate chain branched by C-4 Xyl1. As a continuation of our research on the glycosides from this species, four new chitonoidosides, E_1_ (**1**), F (**2**), G (**3**), and H (**4**), are reported. The chemical structures of **1**–**4** were established through the analyses of the ^1^H, ^13^C NMR, 1D TOCSY, and 2D NMR (^1^H, ^1^H-COSY, HMBC, HSQC, and ROESY) spectra as well as HR-ESI mass spectra. All the original spectra are presented in Appendix A. The hemolytic activity against human erythrocytes and cytotoxic activities against human adenocarcinoma HeLa, colorectal adenocarcinoma DLD-1, and monocyte THP-1 cells were examined.

## 2. Results and Discussion

### 2.1. Structural Elucidation of the Glycosides

The concentrated ethanolic extract of the sea cucumber *Psolus chitonoides* was submitted to hydrophobic chromatography on a Polychrom-1 column (powdered Teflon, Biolar, Latvia). The glycosides were eluted after washing with water as a mobile phase to eliminate salts and inorganic impurities with 50% EtOH. The obtained glycoside fraction was separated by the chromatography on Si gel columns with the stepped gradient of eluents CHCl3/EtOH/H2O (100:75:10), (100:100:17), and (100:125:25) to give the fractions (I–IV). The individual compounds **1**–**4** (Figure 1) were isolated by HPLC of the fractions III and IV on a silica-based column, Supelcosil LC-Si (4.6 × 150 mm) and reversed-phase semipreparative column Supelco Ascentis RP-Amide (10 × 250 mm).

The configurations of the monosaccharide residues in glycosides **1**–**4** were assigned as *D* based on their biogenetic analogies with all other known sea cucumber triterpene glycosides.

It was found that chitonoidosides E_1_ (**1**), F (**2**) and H (**4**) are characterized by holotoxinogenin as aglycone, which was first found in *Apostichopus japonicus* and is broadly distributed in sea cucumber glycosides [16]. This was deduced from the analyses of their ^1^H and ^13^C NMR spectra (Appendix A), which coincided with each other as well as with those of the aglycones of chitonoidosides A_1_, C, and D isolated earlier from the same species—*P. chitonoides* [9].

The molecular formula of chitonoidoside E_1_ (**1**) was determined to be C_65_H_100_O_36_S_2_Na_2_ from the [M_2Na_–Na]^−^ ion peak at *m/z* 1543.5315 (calc. 1543.5339) and [M_2Na_–2Na]^2−^ at *m/z* 760.2734 (calc. 760.2723) in the (−)HR-ESI-MS (Appendix A). The ^1^H and ^13^C NMR spectra of the carbohydrate chain of chitonoidoside E_1_ (**1**) (Table 1, Appendix A) were coincident with those for chitonoidoside E, isolated recently [15] and demonstrated six characteristic doublets of anomeric protons at δ_H_ 4.67–5.11 (*J* = 7.1–8.0 Hz) and six signals of anomeric carbons at δ_C_ 102.3–105.2. The analysis of the ^1^H,^1^H-COSY, 1D TOCSY, HSQC and ROESY spectra of **1** resulted in the assignment of the signals of two xylose residues, one quinovose, one glucose and 3-*O*-methylglucose, as well as 3-*O*-methylxylose residues. The positions of the sulfate groups were determined based on the deshielding, due to α-shifting effect, of the sulfate group’s doubled signal at δ_C_ 67.1, which is characteristic of glucopyranose units sulfated by C-6 (C-6 signals of non-sulfated glucopyranose residues are usually observed at ~δ_C_ 61.2). The signals of C-5 MeGlc4 and C-5 Glc5 were shielded due to the β-effect of the sulfate groups to δ_C_ 75.3 and 75.5, respectively. These data indicate the presence of a hexasaccharide chain with two sulfate groups attached to C-6 MeGlc4 and to C-6 Glc5, and 3-*O*-methylxylose residue as the sixth sugar unit in chitonoidoside E_1_ (**1**). The sequence of monosaccharides and the positions of the glycosidic bonds were confirmed by the correlations H-1 Xyl1/H-3 (C-3) of the aglycone, H-1 Qui2/H-2 (C-2) Xyl1, H-1 Xyl3/H-4 (C-4) Qui2, H-1 MeGlc4/H-3 (C-3) Xyl3, H-1 Glc5/H-4 (C-4) Xyl1, and H-1 MeXyl6/H-3 (C-3) Glc5 in the ROESY and HMBC spectra of **1**, respectively (Table 1, Appendix A).

The (*−*)ESI-MS/MS of **1** (Appendix A) demonstrated the fragmentation of [M_2Na_–Na]^−^ ion at *m/z* 1543.5 with the ion-peaks observed at *m/z* 1423.6 [M_2Na_–Na−NaHSO_4_]^−^, 1266.5 [M_2Na_–Na−MeGlcOSO_3_Na+H]^−^, 1133.5 [M_2Na_–Na−MeGlcOSO_3_Na−Xyl+H]^−^, 987.4 [M_2Na_–Na−MeGlcOSO_3_Na−Xyl−Qui+H]^−^, 841.4 [M_2Na_–Na−MeGlcOSO_3_Na−Xyl−−Qui−MeXyl+2H]^−^, 665.1 [M_2Na_–Na–Agl−MeXyl−GlcOSO_3_Na+H]^−^, 533.1 [M_2Na_–Na–Agl−MeXyl−GlcOSO_3_Na−Xyl+H]^−^, 387.0 [M_2Na_–Na–Agl−MeXyl−GlcOSO_3_Na−Xyl−Qui+H]^−^, 255.0 [M_2Na_–Na–Agl−MeXyl−GlcOSO_3_Na−Xyl−Qui−Xyl+H]^−^. All these ion peaks corroborated the sequence of monosaccharides and the aglycone structure of **1**.

Therefore, chitonoidosides E [15] and E_1_ (**1**) share the identical oligosaccharide moiety (they belong to the same group of glycosides) and differ from each other through the presence/absence of a carbonyl group at C-18 in the aglycones (the difference of their exact masses by 14 *amu* in (−) HR-ESI-MS confirmed this). These data indicate that chitonoidoside E_1_ (**1**) is 3*β*-*O*-{6-*O*-sodium sulfate-3-*O*-methyl-*β*-D-glucopyranosyl-(1→3)-*β*-D-xylopyranosyl-(1→4)-*β*-D-quinovopyranosyl-(1→2)-[3-*O*-methyl-*β*-D-xylopyranosyl-(1→3)-6-*O*-sodium sulfate-*β*-D-glucopyranosyl-(1→4)]-*β*-D-xylopyranosyl}-16-oxo-holosta-9(11),25(26)-diene.

The molecular formula of chitonoidoside F (**2**) was determined to be C54H82O28S_2_Na_2_ from the [M_2Na_–Na]^−^ ion peak at *m/z* 1265.4337 (calc. 1265.4337) and [M_2Na_–2Na]^2−^ ion peak at *m/z* 621.2244 (calc. 621.2269) in the (−)HR-ESI-MS (Appendix A). The 1H NMR spectrum of the carbohydrate part of chitonoidoside F (**2**) exhibited four characteristic doublets at δ_H_ 4.67–5.18 (*J* = 7.6–8.1 Hz), correlated by the HSQC spectrum with corresponding anomeric carbon signals at δ_C_ 102.2–104.9. These signals were indicative of a tetrasaccharide chain with *β*-configurations of glycosidic bonds (Table 2, Appendix A).

An isolated spin system from each monosaccharide residue was analyzed using the ^1^H,^1^H-COSY and 1D TOCSY spectra. Further analysis of the HSQC, ROESY and HMBC spectra resulted in the assignment of all monosaccharide NMR signals. Using this algorithm, the monosaccharides composing the carbohydrate moiety of chitonoidoside F (**2**) were found to be xylose (Xyl1), quinovose (Qui2), glucose (Glc3), and 3-*O*-methylglucose (MeGlc4). The monosaccharide compositions of the other glycosides, reported herein, were established in the same manner.

The signal of C-6 Glc3 in the ^13^C NMR spectrum of **2** was deshielded to δ_C_ 67.2, which is characteristic of sulfation at this position. The signal of C-6 MeGlc4 was observed at δ_C_ 61.2, indicating the absence of a sulfate group at this position, although the MS data indicated the presence of two sulfate groups in **2**. The signal of C-4 MeGlc4 was deshielded to δ_C_ 76.3 when compared to the same signal of terminal 3-*O*-methylglucose residues of the glycosides lacking a sulfate group (δ_C_ ~70.0) [12]. Moreover, the signals of C-3 and C-5 MeGlc4 in the spectrum of **2** were shielded to δ_C_ 85.3 and 76.4, respectively, due to the *β*-shifting effect of a sulfate group. Therefore, the 3-*O*-methylglucose residue in the sugar part of **2** was sulfated by C-4. This structural feature was only found once in previous research—in the glycosides of *Colochirus quadrangularis* [17]. The observation of 3-*O*-methylxylose residue sulfated by C-4 in the glycosides of *P. chitonoides* [15] clearly demonstrates the presence of a specific sulfatase capable of attaching a sulfate group to C-4 of monosaccharides in pyranose form.

The positions of glycosidic linkages, established by the ROESY and HMBC spectra of **2** revealed the uncommon architecture of the sugar chain with disaccharide fragment 4-*O*-sodium sulfate-3-*O*-methyl-*β*-D-glucopyranosyl-(1→3)-6-*O*-sodium sulfate-*β*-D-glucopyranosyl-(1→4) attached to the first (Xyl1) residue, while quinovose was a terminal unit in the reduced bottom semi-chain (Table 2). Two tetraosides whose carbohydrate part featured the same architecture were found only in the sea cucumber *Thyonidium kurilensis* [18].

The (*−*)ESI-MS/MS of **2** (Appendix A) demonstrated the fragmentation of [M_2Na_–Na]^−^ ion at *m/z* 1265.4 resulting in the ion-peaks appearance at *m/z* 1146.5 [M_2Na_–Na−NaSO_4_]^−^, 987.4 [M_2Na_–Na−MeGlcOSO_3_Na+H]^−^, 841.4, [M_2Na_–Na−MeGlcOSO_3_Na−Qui)+H]^−^, corroborating the notion that sulfated 3-*O*-methylglucose and quinovose are terminal monosaccharides. All these data indicate that chitonoidoside F (**2**) is 3*β*-*O*-{*β*-D-quinovopyranosyl-(1→2)-(4-*O*-sodium sulfate-3-*O*-methyl-*β*-D-glucopyranosyl-(1→3)-6-*O*-sodium sulfate-*β*-D-glucopyranosyl-(1→4))-*β*-D-xylopyranosyl}-16-oxo-holosta-9(11),25(26)-diene.

The molecular formula of chitonoidoside G (**3**) was determined to be C_66_H_104_O_36_S_2_Na_2_ from the [M_2Na_–Na]^−^ ion peak at *m/z* 1559.5646 (calc. 1559.5652) and [M_2Na_–2Na]^2−^ ion peak at *m/z* 768.2895 (calc. 768.2880) in the (*−*)HR-ESI-MS (Appendix A).

Based on the absence of the signals of 18(20)-lactone at δ_C_ ~178 (C-18) and ~83 (C-20) in the ^13^C NMR spectrum of **3**, the aglycone with 18(20)-ether bond instead of the lactone was supposed to be present. The NMR spectra of the aglycone part of chitonoidosides G (**3**) and A [15], where this aglycone was first found, were almost coincident with each other (Table 3, Appendix A).

The ^1^H and ^13^C NMR spectra of the carbohydrate chain of chitonoidoside G (**3**) (Table 4, Appendix A) demonstrated six signals of anomeric protons at δ_H_ 4.66–5.18 (d, *J* = 6.9–7.9 Hz), corresponding to the signals of the anomeric carbons at δ_C_ 102.2–104.8. These signals indicated the presence of a hexasaccharide moiety with *β*-glycosidic bonds. The monosaccharide composition of **3** was determined as two xyloses (Xyl1 and Xyl3), quinovose (Qui2), glucose (Glc5), and two 3-*O*-methylglucoses (MeGlc4 and MeGlc6). The doubled signal at δ_C_ 67.1 indicated the presence of two sulfate groups. Using the 1H,1H-COSY and 1D TOCSY spectra their positions were deduced as C-6 MeGlc4 and C-6 Glc5. The comparison of the ^13^C NMR spectra of the carbohydrate parts of the chitonoidosides G (**3**) and E_1_ (**1**) demonstrated the closeness of the signals of the monosaccharide units from first to fifth. The signals of the terminal sixth monosaccharide residue in **3** were assigned as 3-*O*-methylglucose instead of 3-*O*-methylxylose in **1**. The sequence of monosaccharides and the positions of the glycosidic bonds were confirmed by the correlations in the ROESY and HMBC spectra of **3** (Table 4, Appendix A). Therefore, chitonoidoside G (**3**) is the first compound in the combinatorial library of the glycosides from *P. chitonoides* to feature non-sulfated 3-*O*-methylglucose as terminal residue in the upper semi-chain.

The (*−*)ESI-MS/MS of **3** (Appendix A) demonstrated the fragmentation of [M_2Na_–Na]^−^ ion at *m/z* 1559.6 leading to the ion-peaks appearance at *m/z* 1439.6 [M_2Na_–Na−NaHSO_4_]^−^, 1383.6 [M_2Na_–Na−MeGlc+H]^−^, 1281.6 [M_2Na_−Na−MeGlcOSO_3_Na+H]^−^, 1149.5 [M_2Na_–Na−MeGlcOSO_3_Na−Xyl+H]^−^, 1003.5 [M_2Na_–Na−MeGlcOSO_3_Na−Xyl−Qui+H]^−^, 533.1 [M_2Na_–Na–Agl−MeGlc−MeGlcOSO_3_Na−Xyl+H]^−^, 387.0 [M_2Na_–Na–Agl−MeGlc−MeGlcOSO_3_Na−Xyl−Qui+H]^−^, 255.0 [M_2Na_–Na–Agl−MeGlc−MeGlcOSO_3_Na−Xyl−Qui−Xyl+H]^−^, confirming the sequence of monosaccharides and the aglycone structure of **3**.

These data indicate that chitonoidoside G (**3**) is 3*β*-*O*-{6-*O*-sodium sulfate-3-*O*-methyl-*β*-D-glucopyranosyl-(1→3)-*β*-D-xylopyranosyl-(1→4)-*β*-D-quinovopyranosyl-(1→2)-[3-*O*-methyl-*β*-D-glucopyranosyl-(1→3)-6-*O*-sodium sulfate-*β*-D-glucopyranosyl-(1→4)]-*β*-D-xylopyranosyl}-16-oxo-18(20)-epoxylanosta-9(11),25(26)-diene.

The molecular formula of chitonoidoside H (**4**) was determined to be C_59_H_90_O_32_S_2_Na_2_ from the [M_2Na_–Na]^−^ ion peak at *m/z* 1397.4749 (calc. 1397.4760) and [M_2Na_–2Na]^2−^ ion peak at *m/z* 687.2442 (calc. 687.2434) in the (*−*)HR-ESI-MS (Appendix A).

The ^1^H and ^13^C NMR spectra of the carbohydrate chain of chitonoidoside H (**4**) (Table 5, Appendix A) demonstrated five signals of anomeric protons at δ_H_ 4.66–5.18 (d, *J* = 6.9–8.1 Hz), corresponding to the signals of the anomeric carbons at δ_C_ 102.3–104.8, which indicated the presence of a pentasaccharide oligosaccharide chain with *β*-glycosidic bonds between the monosaccharides. The analysis of the 1H,1H-COSY, 1D, and 2D TOCSY and HSQC spectra resulted in the assignment of the signals of two xyloses (Xyl1 and Xyl3), quinovose (Qui2), glucose (Glc4), and 3-*O*-methylglucose (MeGlc5). One sulfate group in chitonoidoside H (**4**) was attached to C-6 Glc4 (in the upper semi-chain), which is a typical position for chitonoidosides that feature a branch point in the sugar chain at C-4 Xyl1, with the exception of chitonoidoside B [15], containing one sulfate group. The position of the second sulfate group was determined as C-4 MeGlc5, based on the deshielding of the signal of C-4 MeGlc5 to δ_C_ 76.2 (α-shifting effect of sulfate group) and the shielding of the signals C-3 and C-5 MeGlc5 to δ_C_ 85.2 and 76.4 (β-shifting effect of sulfate group), respectively, compared to the signals in the chitonoidoside G (**3**). The sequence of the monosaccharides and the positions of the glycosidic bonds deduced by the ROESY and HMBC spectra of **4** displayed the bottom semi-chain, composed of two monosaccharide units, and the upper semi-chain, composed of three monosaccharide units, forming a chain with an uncommon architecture (Table 5, Appendix A).

The comparison of the ^13^C NMR spectra of the carbohydrate parts of **4** and **2** displayed their difference only in the presence of the signals of additional xylose residue (in the bottom semi-chain of **4**), as well as the glycosylation effect at C-4 Qui2, whose signal was observed at δ_C_ 85.6, instead of δ_C_ 76.2 (C-4 Qui2), observed in **2**. Therefore, the chitonoidosides F (**2**) and H (**4**) can be considered as sequential steps in the biosynthesis of the glycosides in *P. chitonoides*.

The (*−*)ESI-MS/MS of **4** (Appendix A) demonstrated the fragmentation of [M_2Na_–Na]^−^ ion at *m/z* 1397.5 resulted in the fragmentary ion-peaks at *m/z* 1277.5 [M_2Na_–Na−NaHSO_4_]^−^, 1119.5 [M_2Na_–Na−MeGlcOSO_3_Na+H]^−^, 1119.5 [M_2Na_–Na−Xyl−Qui]^−^, 987.4 [M_2Na_–Na−MeGlcOSO_3_Na−Xyl+H]^−^, 841.4 [M_2Na_–Na−MeGlcOSO_3_Na−Xyl−Qui+H]^−^, 519.0 [M_2Na_–Na–Agl−MeGlcOSO_3_Na−Xyl+H]^−^, 399.0 [M_2Na_–Na–Agl−MeGlcOSO_3_Na−Xyl−NaHSO_4_]^−^, 373.0 [M_2Na_–Na–Agl−Xyl−Qui−MeGlcOSO_3_Na+H]^−^, 255.0 [M_2Na_–Na–Agl−MeGlcOSO_3_Na−GlcOSO_3_Na–Xyl+H]^−^, confirming both the sequence of monosaccharides and the aglycone structure of **4**.

These data indicate that chitonoidoside H (**4**) is 3*β*-*O*-{*β*-D-xylopyranosyl-(1→4)-*β*-D-quinovopyranosyl-(1→2)-[4-*O*-sodium sulfate-3-*O*-methyl-*β*-D-glucopyranosyl-(1→3)-6-*O*-sodium sulfate-*β*-D-glucopyranosyl-(1→4)]-*β*-D-xylopyranosyl}-16-oxo-holosta-9(11),25(26)-diene.

### 2.2. Bioactivity of the Glycosides

The cytotoxic activity of sea cucumber glycosides against different cell types and cell lines, including HeLa and THP-1, has been extensively studied [1,19]. This has led to deeper understanding of the mechanisms of the anticancer activities of glycosides [7,8,20]. Different tumor cell lines exhibit different sensitivities to the cytotoxic effects of sea cucumber glycosides, depending on their chemical structures. This can be of special interest for the development of therapy for certain types of cancer [21].

The cytotoxic activities of compounds **1**–**4** and chitonoidoside E against human erythrocytes and the human cancer cell lines, adenocarcinoma HeLa, colorectal adenocarcinoma DLD-1, monocytes THP-1, and leukemia promyeloblast HL-60 were investigated (Table 6). The previously tested chitonoidoside A [9] and cisplatin were used as the positive controls.

All the tested compounds demonstrated strong hemolytic activity, with hexaosides **1**, **3** and chitonoidoside E proving to be the most active. The human erythrocytes were more sensitive to the membranolytic action of tested compounds (Table 6) compared to the cancer cells, which is similar to previous data concerning mouse erythrocytes [12,13,15,17]. The monocyte cell line THP-1 and the erythrocytes were comparably sensitive to the action of the glycosides.

A similar tendency for **1**, **3**, and chitonoidoside E was observed for the cytotoxicity against the adenocarcinoma HeLa and HL-60 cells, but the demonstrated effects were moderate. Chitonoidoside E possessing a hexasaccharide chain and the aglycone with a 18(20)-ether bond was the most active; the activity of hexaosides **1** and **3**, which differed through the sixth monosaccharide residue, and the activity of pentaoside **4,** were close to each other against erythrocytes and HeLa and HL-60 cells, but significantly differed in relation to DLD-1 and THP-1 cells. The tetraoside with a shortened bottom semi-chain, chitonoidoside F (**2**), exhibited the weakest membranolytic effect in the series. The most significant difference in the activity of **2** and the other compounds of the series was observed for the DLD-1 and HL-60 cell lines, confirming the diversity in the sensitivities of the cancer cell lines. The presence of the aglycone with an 18(20)-ether bond (in **3**) did not decrease the activity of the glycosides. The latter two peculiarities are in good accordance with the biologic activity of a previously analyzed series of chitonoidosides—the glycosides of *P. chitonoides* [15].

### 2.3. Metabolic Network of Carbohydrate Chains of Chitonoidosides of the Groups A–H

The aglycones and carbohydrate chains of triterpene glycosides are biosynthesized simultaneously and independently from each other [12,22,23], leading to the tremendous structural diversity in this class of natural products from sea cucumbers. Glycosides with identical to sugar moieties that differ in their aglycone structures are considered to belong to a group of glycosides named by particular letter (some groups may consist of one compound). Thus, eight groups of chitonoidosides, A–H, were discovered in *P. chitonoides*, including three types of tetraosides (groups A, C, and F), differing in their architecture and monosaccharide composition, two types of pentaosides (groups D and H) with analogical differences, and three types of hexaosides (groups B, E, and G), differing in their terminal monosaccharide residues and the positions of their sulfate groups. The biosynthesis of sugar moieties occurs through the sequential connection of monosaccharides to certain positions, forming carbohydrate chains. The first branchpoint in the biosynthesis of the chitonoidosides of diverse groups is the attachment of the third sugar residue to the bioside consisting of Xyl1 and Qui2. When the third sugar (xylose) attaches to C-4 Qui2, with the subsequent attachment of 3-OMeGlc and sulfation, the growth of the sugar chain ends with the formation of tetraosides, namely the chitonoidosides of group A (such chains are considered as linear). The attachment of glucose as the third sugar unit to C-4 Xyl1, followed by the addition of glycosylation with different monosaccharide residues (3-*O*-methylxylose or 3-*O*-methylglucose) and sulfation, leads to the formation of the chitonoidosides of groups C and F (Figure 2). The elongation of the bottom semi-chain in chitonoidoside F (by C-4 Qui2) results in chitonoidoside H formation. The further elongation of the sugar chains of the chitonoidosides belonging to the group A and additional sulfation lead to the formation of chitonoidosides of the group D, followed by the subsequent formation of chitonoidosides belonging to group G. When the elongation of the sugar chains of the chitonoidosides of group A occurs without sulfation, the chitonoidosides of the group B are biosynthesized.

Noticeably, that the sulfation of terminal monosaccharide in the upper semi-chain of tetraosides—the chitonoidosides of groups C and F—precedes the subsequent elongation of the bottom semi-chain. Therefore, these glycosides cannot be the precursors of the chitonoidosides of group E containing non-sulfated terminal 3-*O*-methylxylose in the upper semi-chain. Similarly, hexaosides—chitonoidosides of group B—with the only sulfate group at C-6 MeGlc4 cannot be formed from pentaosides, the chitonoidosides of group D because, of the sulfation of Glc5 in the latter. Additionally, the carbohydrate chain of the chitonoidosides of group E can be formed through the sulfation of the chain of chitonoidosides of group B or through the glycosylation of the chain of chitonoidosides of group D. All these data demonstrate that glycosylation and sulfation are competitive and parallel/simultaneous processes in carbohydrate chain biosynthesis.

## 3. Materials and Methods

### 3.1. General Experimental Procedures

Specific rotation, PerkinElmer 343 Polarimeter (PerkinElmer, Waltham, MA, USA); NMR, Bruker AMX 500 (Bruker BioSpin GmbH, Rheinstetten, Germany) (500.12/125.67 MHz (^1^H/^13^C) spectrometer; ESI MS (positive and negative ion modes), Agilent 6510 Q-TOF apparatus (Agilent Technology, Santa Clara, CA, USA), sample concentration 0.01 mg/mL; HPLC, Agilent 1260 Infinity II with a differential refractometer (Agilent Technology, Santa Clara, CA, USA); columns Supelcosil LC-Si (4.6 × 150 mm, 5 µm) and Ascentis RP-Amide (10 × 250 mm, 5 µm) (Supelco, Bellefonte, PA, USA).

### 3.2. Animals and Cells

The specimens of the holothurian *Psolus chitonoides* (family Psolidae; order Dendrochi-rotida) were harvested in the Bering Sea during the 14th expedition cruise on board the r/v “Akademik Oparin” on August 24, 1991, north of Bering Island (Commander Islands). The harvesting was carried out by the Sigsbee trawl at depths of 100–150 m. The animals were taxonomically determined by Dr. Alexey V. Smirnov, Zoological Institute of the Russian Academy of Sciences. Voucher specimens are kept at the Zoological Institute of RAS, St. Petersburg, Russia.

The human erythrocytes were purchased from the Station of Blood Transfusion in Vladivostok. The human adenocarcinoma cell line HeLa cells were provided by the N.N. Blokhin National Medicinal Research Center of Oncology of the Ministry of Health Care of the Russian Federation, (Moscow, Russia). The human colorectal adenocarcinoma cell line DLD-1 CCL-221™ cells and the human monocytes THP-1 TIB-202TM, as well as the human promyeloblast cell line HL-60 CCL-240, were received from ATCC (Manassas, VA, USA). The HeLa cell line was cultured in the medium of DMEM (Gibco Dulbecco’s Modified Eagle Medium), with a 1% penicillin/streptomycin sulfate (Biolot, St. Petersburg, Russia) and 10% fetal bovine serum (FBS) (Biolot, St. Petersburg, Russia). The cells from the DLD-1, HL-60, and THP-1 lines were cultured in an RPMI medium composed of 1% penicillin/streptomycin (Biolot, St. Petersburg, Russia) and 10% fetal bovine serum (FBS) (Biolot, St. Petersburg, Russia). All the cells were incubated at 37 °C in a humidified atmosphere at 5% (*v*/*v*) CO_2_.

The study was conducted according to the guidelines of the Declaration of Helsinki, and approved by the Ethics Committee of the Pacific Institute of Bioorganic Chemistry (Protocol No. 0037.12.03.2021).

### 3.3. Extraction and Isolation

The sea cucumbers were minced and kept in EtOH at +10 °C. Next, they were extracted twice with refluxing 60% EtOH. The combined extracts were concentrated to dryness in a vacuum, dissolved in H2O, and chromatographed on a Polychrom-1 column (powdered Teflon, Biolar, Latvia). Eluting first the inorganic salts and impurities with H2O and then the glycosides with 50% EtOH produced 3200 mg of crude glycoside fraction, which was submitted to stepwise column chromatography on Si gel using CHCl3/EtOH/H2O (100:75:10), (100:100:17) and (100:125:25) as mobile phases to produce fractions I–IV. The HPLC of fraction III on the silica-based column Supelcosil LC-Si (4.6 × 150 mm, 5 µm) with CHCl_3_/MeOH/H_2_O (60/25/4) as the mobile phase resulted in the isolation of six subfractions (III.1–III.6). The subsequent HPLC of subfraction III.4 on Supelco Ascentis RP-Amide (10 × 250 mm) with CH_3_CN/H_2_O/NH_4_OAc (1 M water solution) (40/59/1) as the mobile phase resulted in the isolation of seven fractions (III.4.1–III.4.7). Repeated chromatography of III.4.3 on the same column but with MeOH/H_2_O/NH_4_OAc (1 M water solution) (67/31/2) as the mobile phase produced 9.2 mg of chitonoidoside E_1_ (**1**). Rechromatography of III.4.6 using mobile phase MeOH/H_2_O/NH_4_OAc (1 M water solution) (68/30/2) produced 4.3 mg of chitonoidoside F (**2**). Fraction IV obtained after Si gel column chromatography was submitted to HPLC on Supelcosil LC-Si (4.6 × 150 mm, 5 µm) with CHCl_3_/MeOH/H_2_O (63/27/4) as the mobile phase to produce a set of subfractions (IV.1–IV.6). Chitonoidoside G (**3**) (3.7 mg) was isolated as a result of HPLC of subfraction IV.3 on Supelco Ascentis RP-Amide (10 × 250 mm) with MeOH/H_2_O/NH_4_OAc (1 M water solution) (72/26/2) as the mobile phase. Chitonoidoside H (**4**) (4.7 mg) was obtained after HPLC of subfraction IV.1 on Supelco Ascentis RP-Amide (10 × 250 mm) with MeOH/H_2_O/NH_4_OAc (1 M water solution) (70/28/2) as the mobile phase.

#### 3.3.1. Chitonoidoside E_1_ (1)

Colorless powder; (α)_D_^20^−35° (*c* 0.1, 50% MeOH). NMR: See Appendix A and Table 1, Appendix A. (*−*)HR-ESI-MS *m/z*: 1543.5315 (calc. 1543.5339) [M_2Na_*−*Na]^−^, 760.2734 (calc. 760.2723) [M_2Na_–2Na]^2−^. (*−*)ESI-MS/MS *m/z*: 1423.6 [M_2Na_–Na−NaHSO_4_]^−^, 1266.5 [M_2Na_–Na−C_7_H_12_O_8_SNa (MeGlcOSO_3_Na)+H]^−^, 1133.5 [M_2Na_–Na−C_7_H_12_O_8_SNa (MeGlcOSO_3_Na)−C_5_H_8_O_4_ (Xyl)+H]^−^, 987.4 [M_2Na_–Na−C_7_H_12_O_8_SNa (MeGlcOSO_3_Na)−C_5_H_8_O_4_ (Xyl)−C_6_H_10_O_4_ (Qui)+H]^−^, 841.4 [M_2Na_–Na−C_7_H_12_O_8_SNa (MeGlcOSO_3_Na)−C_5_H_8_O_4_ (Xyl)−C_6_H_10_O_4_ (Qui)−C_6_H_11_O_4_ (MeXyl)+2H]^−^, 665.1 [M_2Na_–Na–C_30_H_44_O_4_ (Agl)−C_6_H_11_O_4_ (MeXyl)−C_6_H_9_O_8_SNa (GlcOSO_3_Na)+H]^−^, 533.1 [M_2Na_–Na–C_30_H_44_O_4_ (Agl)−C_6_H_11_O_4_ (MeXyl)−C_6_H_9_O_8_SNa (GlcOSO_3_Na)−C_5_H_8_O_4_ (Xyl)+H]^−^, 387.0 [M_2Na_–Na–C_30_H_44_O_4_ (Agl)−C_6_H_11_O_4_ (MeXyl)−C_6_H_9_O_8_SNa (GlcOSO_3_Na)−C_5_H_8_O_4_ (Xyl)−C_6_H_10_O_4_ (Qui)+H]^−^, 255.0 [M_2Na_–Na–C_30_H_44_O_4_ (Agl)−C_6_H_11_O_4_ (MeXyl)−C_6_H_9_O_8_SNa (GlcOSO_3_Na)−C_5_H_8_O_4_ (Xyl)−C_6_H_10_O_4_ (Qui)−C_5_H_8_O_4_ (Xyl)+H]^−^ (Appendix A).

#### 3.3.2. Chitonoidoside F (2)

Colorless powder; (α)_D_^20^*−*19° (*c* 0.1, 50% MeOH). NMR: See Appendix A and Table 2, Appendix A. (*−*)HR-ESI-MS *m/z*: 1265.4337 (calc. 1265.4337) [M_2Na_–Na)^−^, 621.2244 (calc. 621.2269) (M_2Na_–2Na]^2−^; (*−*)ESI-MS/MS *m/z*: 1146.5 [M_2Na_–Na−NaSO_4_]^−^, 987.4 [M_2Na_–Na−C_7_H_12_O_8_SNa (MeGlcOSO_3_Na)+H]^−^, 841.4, [M_2Na_–Na−C_7_H_12_O_8_SNa (MeGlcOSO_3_Na)−C_6_H_10_O_4_ (Qui)+H]^−^ (Appendix A).

#### 3.3.3. Chitonoidoside G (3)

Colorless powder; (α)_D_^20^*−*48° (*c* 0.1, 50% MeOH). NMR: See Table 3 and Table 4, Appendix A. (*−*)HR-ESI-MS *m/z*: 1559.5646 (calc. 1559.5652) [M_2Na_–Na]^−^, 768.2895 (calc. 768.2880) [M_2Na_–2Na]^2−^; (*−*)ESI-MS/MS *m/z*: 1439.6 [M_2Na_–Na−NaHSO_4_]^−^, 1383.6 [M_2Na_–Na−C_7_H_13_O_5_ (MeGlc)+H]^−^, 1281.6 [M_2Na_–Na−C_7_H_12_O_8_SNa (MeGlcOSO_3_Na)+H]^−^, 1149.5 [M_2Na_–Na−C_7_H_12_O_8_SNa (MeGlcOSO_3_Na)−C_5_H_8_O_4_ (Xyl)+H]^−^, 1003.5 [M_2Na_–Na−C_7_H_12_O_8_SNa (MeGlcOSO_3_Na)−C_5_H_8_O_4_ (Xyl)−C_6_H_10_O_4_ (Qui)+H]^−^, 533.1 [M_2Na_–Na–C_30_H_46_O_3_ (Agl)−C_7_H_13_O_5_ (MeGlc)−C_7_H_12_O_8_SNa (MeGlcOSO_3_Na)−C_5_H_8_O_4_ (Xyl)+H]^−^, 387.0 [M_2Na_–Na–C_30_H_46_O_3_ (Agl)−C_7_H_13_O_5_ (MeGlc)−C_7_H_12_O_8_SNa (MeGlcOSO_3_Na)−C_5_H_8_O_4_ (Xyl)−C_6_H_10_O_4_ (Qui)+H]^−^, 255.0 [M_2Na_–Na–C_30_H_46_O_3_ (Agl)−C_7_H_13_O_5_ (MeGlc)−C_7_H_12_O_8_SNa (MeGlcOSO_3_Na)−C_5_H_8_O_4_ (Xyl)−C_6_H_10_O_4_ (Qui)−C_5_H_8_O_4_ (Xyl)+H]^−^ (Appendix A).

#### 3.3.4. Chitonoidoside H (4)

Colorless powder; (α)_D_^20^*−*28° (*c* 0.1, 50% MeOH). NMR: See Appendix A and Table 5, Appendix A. (*−*)HR-ESI-MS *m/z*: 1397.4749 (calc. 1397.4760) [M_2Na_–Na]^−^, 687.2442 (calc. 687.2434) [M_2Na_–2Na]^2−^; (*−*)ESI-MS/MS *m/z*: 1277.5 [M_2Na_–Na−NaHSO_4_]^−^, 1119.5 [M_2Na_–Na−C_7_H_12_O_8_SNa (MeGlcOSO_3_Na)+H]^−^, 1119.5 [M_2Na_–Na−C_5_H_8_O_4_ (Xyl) −C_6_H_10_O_4_ (Qui)]^−^, 987.4 [M_2Na_–Na−C_7_H_12_O_8_SNa (MeGlcOSO_3_Na)−C_5_H_8_O_4_ (Xyl)+H]^−^, 841.4 [M_2Na_–Na−C_7_H_12_O_8_SNa (MeGlcOSO_3_Na)−C_5_H_8_O_4_ (Xyl)−C_6_H_10_O_4_ (Qui)+H]^−^, 519.0 [M_2Na_–Na–C_30_H_44_O_4_ (Agl)−C_7_H_12_O_8_SNa (MeGlcOSO_3_Na)−C_5_H_8_O_4_ (Xyl)+H]^−^, 399.0 [M_2Na_–Na–C_30_H_44_O_4_ (Agl)−C_7_H_12_O_8_SNa (MeGlcOSO_3_Na)−C_5_H_8_O_4_ (Xyl)−NaHSO_4_]^−^, 373.0 [M_2Na_–Na–C_30_H_44_O_4_ (Agl)−C_5_H_8_O_4_ (Xyl)−C_6_H_10_O_4_ (Qui)−C_7_H_12_O_8_SNa (MeGlcOSO_3_Na)+H]^−^, 255.0 [M_2Na_–Na–C_30_H_46_O_3_ (Agl)−C_7_H_12_O_8_SNa (MeGlcOSO_3_Na)−C_6_H_9_O_8_SNa (GlcOSO_3_Na)–C_5_H_8_O_4_ (Xyl)+H]^−^ (Appendix A).

### 3.4. Cytotoxic Activity (MTT Assay Applied for HeLa Cells)

All the studied substances (including the chitonoidoside A and cisplatin, used as positive controls) were tested in concentrations from 0.1 µM to 100 µM using twofold dilution in d-H2O. The cell suspension (180 µL) and solutions (20 µL) of the tested compounds in different concentrations were injected in wells of 96 well plates (1 × 104 cells/well) and incubated at 37 °C for 24 h in 5% CO_2_. After incubation, the tested substances with medium were replaced by 100 µL of fresh medium. Next, 10 µL of MTT (3-(4,5-dimethylthiazol-2-yl)-2,5-diphenyltetrazolium bromide) (Sigma-Aldrich, St. Louis, MO, USA) stock solution (5 mg/mL) was added to each well, followed by the incubation of the microplate for 4 h. Subsequently, 100 µL of SDS-HCl solution (1 g SDS/10 mL d-H2O/17 µL 6 N HCl) was added to each well and incubated for 18 h. The absorbance of the converted dye, formazan, was measured with a Multiskan FC microplate photometer (Thermo Fisher Scientific, Waltham, MA, USA) at 570 nm. The cytotoxic activity of the tested compounds was calculated as the concentration that caused 50% cell metabolic activity inhibition (IC50). The experiments were carried out in triplicate, *p* < 0.05.

### 3.5. Cytotoxic Activity (MTS Assay Applied for DLD-1, THP-1 and HL-60 Cells)

The cells of the HL-60 line (6 × 10^3^/200 µL) were placed in 96 well plates at 37 °C for 24 h in a 5% CO_2_ incubator. The cells were treated with tested substances and chitonoidoside A and cisplatin were used as positive controls at concentrations from 0 to 100 µM for an additional 24 h of incubation. Next, the cells were incubated with 10 µL MTS (3-(4,5-dimethylthiazol-2-yl)-5-(3-carboxymethoxyphenyl)-2-(4-sulfophenyl)-2H-tetrazolium) for 4 h, and the absorbance in each well was measured at 490/630 nm with plate reader PHERA star FS (BMG Labtech, Ortenberg, Germany). The experiments were carried out in triplicate and the mean absorbance values were calculated. The results were presented as the percentage of inhibition that produced a reduction in absorbance after the tested compound treatment compared to the non-treated cells (negative control), *p* < 0.01.

### 3.6. Hemolytic Activity

The erythrocytes were isolated from human blood through centrifugation with phosphate-buffered saline (PBS) (pH 7.4) at 4 °C for 5 min by 450× *g* on centrifuge LABOFUGE 400R (Heraeus, Hanau, Germany) three times. Next, the residue of the erythrocytes was resuspended in an ice-cold phosphate saline buffer (pH 7.4) to a final optical density of 1.5 at 700 nm, and kept on ice. For the hemolytic assay, 180 µL of erythrocyte suspension was mixed with 20 µL of test compound solution (including chitonoidoside A used as positive control) in V-bottom 96 well plates. After 1 h of incubation at 37 °C, the plates were exposed to centrifugation for 10 min at 900× *g* on laboratory centrifuge LMC-3000 (Biosan, Riga, Latvia). Next, 100 µL of supernatant was carefully selected and transferred into new flat-plates, respectively. The lysis of the erythrocytes was determined by measuring of the concentration of hemoglobin in the supernatant with microplate photometer Multiskan FC (Thermo Fisher Scientific, Waltham, MA, USA), λ = 570 nm. The effective dose causing 50% hemolysis of erythrocytes (ED50) was calculated using the computer program SigmaPlot 10.0. All the experiments were performed in triplicate, *p* < 0.01.

## 4. Conclusions

The continuation of the research into triterpene glycosides from the sea cucumber *Psolus chitonoides* resulted in the isolation of four previously unknown chitonoidosides E_1_ (**1**), F (**2**), G (**3**) and H (**4**). The compounds characterized by two types of the aglycones (holotoxinogenin and its structural analog with 18(20)-epoxy-cycle instead of a lactone) were also found earlier in the series of chitonoidosides A–E. The compounds **1**–**4** differed in the number of monosaccharides in their sugar chains (from four to six), the architecture of these chains (for tetra- and pentaosides), their monosaccharide composition, and the positions of their sulfate groups. Terminal 3-*O*-methylglucose unit was sulfated by C-4 in chitonoidosides F (**2**) and H (**4**), which is a rare position for a sulfate group from sea cucumber glycosides. Notably, in the first series of the studied glycosides from *P. chitonoides* [15] 3-*O*-methylxylose residue sulfated by C-4 was found. This indicates the presence of specific sulfatase in this species of the sea cucumber, capable of attaching the sulfate group to C-4 of monosaccharides in pyranose form. The observed “structure-activity relationships” were as follows: tetraosides with a shortened bottom semi-chain displayed the weakest membranolytic effect; and the activity of the glycosides with the new-type aglycone with a 18(20)-ether bond instead of 18(20)-lactone was comparable with that of the substances with holostane-type aglycones. Hexaosides and tetraosides with linear carbohydrate chains (having bottom semi-chain) were the most active in the series of the glycosides from *P. chitonoides*. The pathways of the biosynthetic transformations were analyzed based on the structures of eight types of carbohydrate chains found in the glycosides of *P. chitonoides*. The analysis revealed and confirmed that glycosylation and sulfation are parallel and competitive processes. All the discussed data broaden knowledge about structural diversity of triterpene glycosides from the sea cucumbers and help us to the understand the complicated biosynthesis process of this class of metabolites, which is currently a large gap in knowledge for scientists.

## Figures and Tables

**Figure 1 marinedrugs-19-00696-f001:**
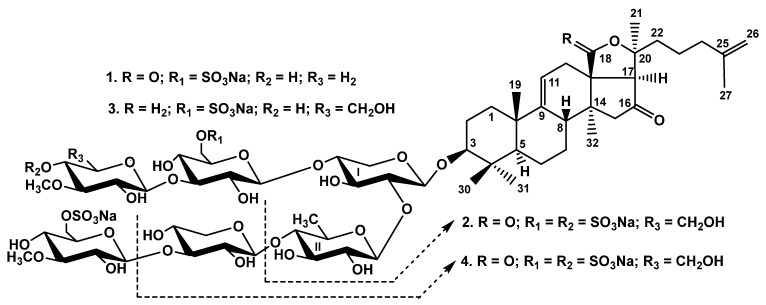
Chemical structures of glycosides isolated from *Psolus chitonoides*: **1**—chitonoidoside E_1_; **2**—chitonoidoside F; **3**—chitonoidoside G; **4**—chitonoidoside H.

**Figure 2 marinedrugs-19-00696-f002:**
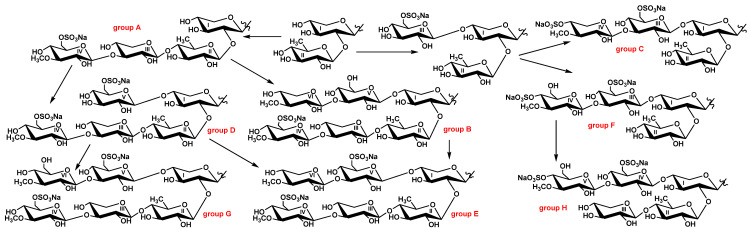
Biosynthetic network of carbohydrate chains of chitonoidosides of the groups A–H.

**Table 1 marinedrugs-19-00696-t001:** ^13^C and ^1^H NMR chemical shifts, HMBC and ROESY correlations of carbohydrate moiety of chitonoidoside E_1_ (**1**).

Atom	δ_C_ Mult. *^a–c^*	δ_H_ Mult. (*J* in Hz) *^d^*	HMBC	ROESY
Xyl1 (1→C-3)				
1	104.8 CH	4.67 d (7.1)	C: 3	H-3; H-3, 5 Xyl1
2	**82.2** CH	3.96 m	C: 1 Qui2	H-1 Qui2
3	75.1 CH	4.16 m		H-1, 5 Xyl1
4	**77.9** CH	4.16 m		H-1 Glc5
5	63.5 CH_2_	4.38 dd (4.3; 12.2)	C: 3 Xyl1	
		3.64 m		H-1 Xyl1
Qui2 (1→2Xyl1)				
1	104.6 CH	5.03 d (8.0)	C: 2 Xyl1	H-2 Xyl1; H-3, 5 Qui2
2	75.7 CH	3.87 t (8.0)	C: 1, 3 Qui2	
3	74.8 CH	3.98 t (9.3)	C: 2 Qui2	H-1 Qui2
4	**85.6** CH	3.49 t (9.3)	C: 1 Xyl3; 3, 5 Qui2	H-1 Xyl3
5	71.4 CH	3.67 dd (6.2; 9.3)		H-1 Qui2
6	17.8 CH_3_	1.61 d (6.2)	C: 4, 5 Qui2	H-4 Qui2
Xyl3 (1→4Qui2)				
1	104.5 CH	4.75 d (7.7)	C: 4 Qui2	H-4 Qui2; H-3, 5 Xyl3
2	73.2 CH	3.84 t (8.5)	C: 1 Xyl3	
3	**87.1** CH	4.04 t (8.5)	C: 1 MeGlc4; 2, 4 Xyl3	H-1 MeGlc4
4	68.7 CH	3.90 m		
5	65.7 CH_2_	4.13 dd (4.3; 11.1)	C: 4 Xyl3	
		3.60 t (11.1)	C: 1, 3 Xyl3	H-1 Xyl3
MeGlc4 (1→3Xyl3)				
1	104.4 CH	5.11 d (7.9)	C: 3 Xyl3	H-3 Xyl3; H-3, 5 MeGlc4
2	74.3 CH	3.80 t (7.9)	C: 1 MeGlc4	
3	86.4 CH	3.64 t (9.1)	C: 2, 4 MeGlc4, OMe	H-1, 5 MeGlc4
4	69.7 CH	3.96 t (9.1)		
5	75.3 CH	4.03 m		H-1, 3 MeGlc4
6	*67.1* CH_2_	4.97 dd (3.0; 11.5)	C: 5 MeGlc4	
		4.71 dd (6.1; 11.5)	C: 5 MeGlc4	
OMe	60.5 CH_3_	3.76 s	C: 3 MeGlc4	
Glc5 (1→4Xyl1)				
1	102.3 CH	4.90 d (7.9)	C: 4 Xyl1	H-4 Xyl1; H-3, 5 Glc5
2	73.2 CH	3.84 t (9.1)	C: 1, 3 Glc5	
3	**86.0** CH	4.07 t (9.1)	C: 1 MeXyl6; 2 Glc5	H-1 MeXyl6; H-1 Glc5
4	68.9 CH	3.88 t (9.1)		
5	75.5 CH	4.05 m		H-1 Glc5
6	*67.1* CH_2_	4.97 dd (1.9; 11.5)	C: 4, 5 Glc5	
		4.70 dd (5.5; 11.5)		
MeXyl6 (1→3Glc5)				
1	105.2 CH	5.07 d (7.9)	C: 3 Glc5	H-3 Glc5; H-3,5 MeXyl6
2	74.2 CH	3.79 t (8.5)	C: 1 MeXyl6	
3	86.6 CH	3.57 t (9.1)	C: 2, 4 MeXyl6; OMe	H-1 MeXyl6; OMe
4	69.9 CH	3.98 t (9.1)	C: 3, 5 MeXyl6	
5	66.4 CH_2_	4.14 dd (5.5; 12.1)	C: 1, 3, 4 MeXyl6	
		3.61 t (11.5)		H-1 MeXyl6
OMe	60.5 CH_3_	3.79 s	C: 3 MeXyl6	H-3 MeXyl6

*^a^* Recorded at 125.67 MHz in C_5_D_5_N/D_2_O (4/1). *^b^* Bold = interglycosidic positions. *^c^* Italic = sulfate position. *^d^* Recorded at 500.12 MHz in C_5_D_5_N/D_2_O (4/1). Multiplicity by 1D TOCSY. The original spectra of **1** are provided in Appendix A.

**Table 2 marinedrugs-19-00696-t002:** ^13^C and ^1^H NMR chemical shifts, HMBC and ROESY correlations of carbohydrate moiety of chitonoidoside F (**2**).

Atom	δ_C_ Mult. *^a–c^*	δ_H_ Mult. (*J* in Hz) *^d^*	HMBC	ROESY
Xyl1 (1→C-3)				
1	104.8 CH	4.67 d (7.6)	C: 3	H-3; H-3, 5 Xyl1
2	**82.1** CH	3.98 t (8.3)	C: 1 Qui2; 3 Xyl1	H-1 Qui2
3	75.1 CH	4.17 t (8.3)	C: 4 Xyl1	
4	**78.0** CH	4.16 m		H-1 Xyl1; H-1 Glc3
5	63.5 CH_2_	4.38 d (10.2)		
		3.63 m		H-1, 3 Xyl1
Qui2 (1→2Xyl1)				
1	104.9 CH	5.06 d (8.1)	C: 2 Xyl1	H-2 Xyl1; H-3, 5 Qui2
2	76.2 CH	3.88 t (9.1)	C: 1 Qui2	H-4 Qui2
3	76.8 CH	4.06 t (9.1)	C: 2, 4 Qui2	H-1, 5 Qui2
4	76.2 CH	3.58 t (9.1)	C: 3, 5 Qui2	H-2 Qui2
5	72.8 CH	3.70 dd (6.3; 9.9)	C: 4 Qui2	H-1, 3 Qui2
6	18.2 CH_3_	1.53 d (6.3)	C: 4, 5 Qui2	
Glc3 (1→4Xyl1)				
1	102.2 CH	4.90 d (8.0)	C: 4 Xyl1	H-4 Xyl1; H-3, 5 Glc3
2	73.3 CH	3.83 t (9.2)	C: 1 Glc3	
3	**86.0** CH	4.17 t (9.2)	C: 2, 4 Glc3; 1 MeGlc4	H-1 MeGlc4; H-1 Glc3
4	68.9 CH	3.87 t (9.2)	C: 5 Glc3	
5	75.1 CH	4.04 m		H-1 Glc3
6	*67.2* CH_2_	4.94 d (10.3)		
		4.68 m		
MeGlc4 (1→3Glc3)				
1	104.3 CH	5.18 d (8.0)	C: 3 Glc3	H-3 Glc3; H-3, 5 MeGlc4
2	74.0 CH	3.86 t (8.7)	C: 1, 3 MeGlc4	H-4 MeGlc4
3	85.3 CH	3.71 t (8.7)	C: 4, 5 MeGlc4, OMe	H-1, 5 MeGlc4; Ome
4	*76.3* CH	4.88 t (9.6)	C: 3, 5 MeGlc4	H-2 MeGlc4
5	76.4 CH	3.86 t (8.7)		H-1 MeGlc4
6	61.2 CH_2_	4.51 d (11.4)		
		4.33 dd (4.4; 11.4)		
OMe	60.7 CH_3_	3.93 s	C: 3 MeGlc4	

*^a^* Recorded at 125.67 MHz in C_5_D_5_N/D_2_O (4/1). *^b^* Bold = interglycosidic positions. *^c^* Italic = sulfate position. *^d^* Recorded at 500.12 MHz in C_5_D_5_N/D_2_O (4/1). Multiplicity by 1D TOCSY. The original spectra of **2** are provided in Appendix A.

**Table 3 marinedrugs-19-00696-t003:** ^13^C and ^1^H NMR chemical shifts, HMBC and ROESY correlations of the aglycone moiety of chitonoidoside G (**3**).

Position	δ_C_ Mult. *^a^*	δ_H_ Mult. (*J* in Hz) *^b^*	HMBC	ROESY
1	35.9 CH_2_	1.60 m		H-11
		1.28 m		
2	26.7 CH_2_	2.08 m		
		1.83 m		H-19, H-30
3	88.7 CH	3.12 dd (4.2; 11.8)	C: 1 Xyl1	H-5, H-31, H1-Xyl1
4	39.6 C			
5	52.7 CH	0.75 brd (12.0)	C: 10, 19	H-3, H-31
6	20.9 CH_2_	1.57 m		H-31
		1.35 m		H-19, H-30
7	28.7 CH_2_	1.57 m		
		1.17 m		
8	40.9 CH	2.31 m		H-18, H-19
9	150.9 C			
10	39.5 C			
11	114.7 CH	5.30 m		H-1
12	33.8 CH_2_	2.38 m		H-32
		2.25 m		H-21
13	56.3 C			
14	40.3 C			
15	50.5 CH_2_	2.47 d (15.9)	C: 14, 16, 32	H-18
		2.19 d (15.9)	C: 13, 16	H-32
16	218.1 C			
17	63.8 CH	2.35 s	C: 12, 13, 16, 18, 20, 21	H-12, H-21, H-22, H-32
18	73.8 CH_2_	4.02 m		
		3.65 d (9.1)	C: 12, 20	
19	22.2 CH_3_	0.97 s	C: 1, 5, 9, 10	H-1, H-2, H-6, H-8, H-18
20	86.6 C			
21	26.1 CH_3_	1.32 s	C: 17, 20, 22	H-12, H-17, H-18, H-22
22	37.8 CH_2_	1.70 m		
		1.57 m		H-21, H-24
23	22.7 CH_2_	1.69 m		
		1.56 m		
24	38.2 CH_2_	1.95 m	C: 23	H-21
25	146.0 C			
26	110.2 CH_2_	4.72 brs	C: 24, 27	
		4.71 brs	C: 24, 27	
27	22.2 CH_3_	1.65 s	C: 24, 25, 26	H-24
30	16.5 CH_3_	0.97 s	C: 3, 4, 5, 31	H-31
31	27.9 CH_3_	1.13 s	C: 3, 4, 5, 30	H-1, H-3, H-5, H-30
32	21.4 CH_3_	0.78 s	C: 13, 14, 15	H-15, H-17

*^a^* Recorded at 125.67 MHz in C_5_D_5_N/D_2_O (4/1). *^b^* Recorded at 500.12 MHz in C_5_D_5_N/D_2_O (4/1). The original spectra of **3** are provided in Appendix A.

**Table 4 marinedrugs-19-00696-t004:** ^13^C and ^1^H NMR chemical shifts, HMBC and ROESY correlations of carbohydrate moiety of chitonoidoside G (**3**).

Atom	δ_C_ Mult. *^a–c^*	δ_H_ Mult. (*J* in Hz) *^d^*	HMBC	ROESY
Xyl1 (1→C-3)				
1	104.8 CH	4.66 d (6.9)	C: 3	H-3; H-3, 5 Xyl1
2	**82.1** CH	3.97 t (8.8)	C: 1 Qui2; 1, 3 Xyl1	H-1 Qui2
3	75.1 CH	4.16 t (8.8)	C: 4 Xyl1	
4	**77.8** CH	4.16 m		
5	63.5 CH_2_	4.37 dd (4.1; 11.8)		
		3.62 m		H-1 Xyl1
Qui2 (1→2Xyl1)				
1	104.5 CH	5.04 d (7.3)	C: 2 Xyl1	H-2 Xyl1; H-3, 5 Qui2
2	75.7 CH	3.87 t (9.0)	C: 1, 3 Qui2	H-4 Qui2
3	74.8 CH	3.98 t (9.0)	C: 2, 4 Qui2	H-5 Qui2
4	**85.6** CH	3.49 t (9.0)	C: 1 Xyl3; 3, 5 Qui2	H-1 Xyl3; H-2 Qui2
5	71.4 CH	3.68 dd (6.2; 9.0)		H-1 Qui2
6	17.8 CH_3_	1.62 d (6.2)	C: 4, 5 Qui2	H-4, 5 Qui2
Xyl3 (1→4Qui2)				
1	104.4 CH	4.75 d (7.7)	C: 4 Qui2	H-4 Qui2; H-3, 5 Xyl3
2	73.2 CH	3.84 t (8.3)	C: 1, 3 Xyl3	
3	**87.0** CH	4.04 t (8.3)	C: 1 MeGlc4; 2, 4 Xyl3	H-1 MeGlc4; H-1 Xyl3
4	68.8 CH	3.89 m	C: 5 Xyl3	
5	65.7 CH_2_	4.12 dd (5.3; 11.2)		
		3.59 d (11.2)	C: 1 Xyl3	H-1 Xyl3
MeGlc4 (1→3Xyl3)				
1	104.6 CH	5.12 d (7.9)	C: 3 Xyl3	H-3 Xyl3; H-3, 5 MeGlc4
2	74.3 CH	3.80 t (8.5)	C: 1 MeGlc4	
3	86.4 CH	3.64 t (8.5)	C: 4 MeGlc4, OMe	H-1 MeGlc4; OMe
4	69.9 CH	3.96 t (8.5)	C: 3, 5, 6 MeGlc4	H-2, 6 MeGlc4
5	75.5 CH	4.03 m		H-1, 3 MeGlc4
6	*67.1* CH_2_	4.97 d (10.7)		
		4.71 dd (5.6; 11.3)	C: 5 MeGlc4	
OMe	60.5 CH_3_	3.76 s	C: 3 MeGlc4	
Glc5 (1→4Xyl1)				
1	102.2 CH	4.88 d (7.9)	C: 4 Xyl1	H-4 Xyl1; H-3, 5 Glc5
2	73.2 CH	3.84 t (9.0)	C: 1, 3 Glc5	
3	**86.0** CH	4.16 t (9.0)	C: 1 MeGlc6; 2 Glc5	H-1 MeGlc6; H-1 Glc5
4	69.0 CH	3.89 t (9.0)	C: 3 Glc5	
5	75.5 CH	4.02 m		H-1 Glc5
6	*67.1* CH_2_	4.93 d (10.7)		
		4.68 dd (6.2; 11.3)		
MeGlc6 (1→3Glc5)				
1	104.4 CH	5.18 d (7.5)	C: 3 Glc5	H-3 Glc5; H-3, 5 MeGlc6
2	74.5 CH	3.84 t (8.8)	C: 1 MeGlc6	
3	86.8 CH	3.66 t (8.8)	C: 2, 4 MeGlc6, OMe	H-1 MeGlc6
4	70.3 CH	3.89 m	C: 5 MeGlc6	H-6 MeGlc6
5	77.5 CH	3.89 m		H-1 MeGlc6
6	61.7 CH_2_	4.34 dd (2.2; 11.7)		
		4.05 dd (5.1; 11.7)	C: 4 MeGlc6	
OMe	60.6 CH_3_	3.80 s	C: 3 MeGlc6	

*^a^* Recorded at 125.67 MHz in C_5_D_5_N/D_2_O (4/1). *^b^* Bold = interglycosidic positions. *^c^* Italic = sulfate position. *^d^* Recorded at 500.12 MHz in C_5_D_5_N/D_2_O (4/1). Multiplicity by 1D TOCSY. The original spectra of **1** are provided in Appendix A.

**Table 5 marinedrugs-19-00696-t005:** ^13^C and ^1^H NMR chemical shifts, HMBC and ROESY correlations of carbohydrate moiety of chitonoidoside H (**4**).

Atom	δ_C_ Mult. *^a–c^*	δ_H_ Mult. (*J* in Hz) *^d^*	HMBC	ROESY
Xyl1 (1→C-3)				
1	104.6 CH	4.66 d (6.9)	C: 3	H-3; H-3, 5 Xyl1
2	**81.9** CH	3.97 m	C: 1 Qui2; 1 Xyl1	H-1 Qui2
3	75.1 CH	4.16 m	C: 4 Xyl1	H-1 Xyl1
4	**78.0** CH	4.15 m		H-1 Glc4
5	63.5 CH_2_	4.38 m	C: 3 Xyl1	
		3.62 m		H-1 Xyl1
Qui2 (1→2Xyl1)				
1	104.4 CH	5.07 d (6.9)	C: 2 Xyl1	H-2 Xyl1; H-5 Qui2
2	75.7 CH	3.87 t (9.2)	C: 1, 3 Qui2	H-4 Qui2
3	74.8 CH	4.00 t (9.2)	C: 2, 4 Qui2	H-1 Qui2
4	**85.6** CH	3.47 t (9.2)	C: 1 Xyl3; 3, 5 Qui2	H-1 Xyl3; H-2 Qui2
5	71.4 CH	3.70 dd (6.2; 9.2)		H-1, 3 Qui2
6	17.8 CH_3_	1.61 d (6.2)	C: 4, 5 Qui2	H-4, 5 Qui2
Xyl3 (1→4Qui2)				
1	104.8 CH	4.70 d (6.6)	C: 4 Qui2	H-4 Qui2; H-3, 5 Xyl3
2	73.3 CH	3.82 t (8.4)	C: 1, 3 Xyl3	
3	77.2 CH	4.05 t (8.4)	C: 2, 4 Xyl3	
4	70.1 CH	4.03 m	C: 3 Xyl3	
5	66.5 CH_2_	4.14 brd (11.2)	C: 3 Xyl3	
		3.59 t (11.2)	C: 1, 3, 4 Xyl3	H-1 Xyl3
Glc4 (1→4Xyl1)				
1	102.3 CH	4.89 d (8.1)	C: 4 Xyl1	H-4 Xyl1; H-3, 5 Glc4
2	74.0 CH	3.82 t (9.1)	C: 1 Glc4	
3	86.0 CH	4.16 t (9.1)	C: 1 MeGlc5	H-1 MeGlc5; H-1 Glc4
4	69.0 CH	3.86 t (9.1)	C: 3 Glc4	
5	75.2 CH	4.03 m		H-1 Glc4
6	*67.2* CH_2_	4.96 d (12.2)		
		4.69 m		
MeGlc5 (1→3Glc4)				
1	104.3 CH	5.18 d (8.1)	C: 3 Glc4	H-3 Glc4; H-3, 5 MeGlc5
2	74.1 CH	3.86 t (9.1)	C: 1 MeGlc5	H-4 MeGlc5
3	85.2 CH	3.71 t (9.1)	C: 4 MeGlc5; OMe	H-1 MeGlc5
4	*76.2* CH	4.88 t (9.1)	C: 3, 5 MeGlc5	H-6 MeGlc5
5	76.4 CH	3.84 t (9.1)		
6	61.7 CH_2_	4.50 d (12.2)		
		4.33 dd (6.2; 12.2)		
OMe	60.7 CH_3_	3.93 s	C: 3 MeGlc5	

*^a^* Recorded at 125.67 MHz in C_5_D_5_N/D_2_O (4/1). *^b^* Bold = interglycosidic positions. *^c^* Italic = sulfate position. *^d^* Recorded at 500.12 MHz in C_5_D_5_N/D_2_O (4/1). Multiplicity by ^1^H, and 1D TOCSY. The original spectra of **1** are provided in Appendix A.

**Table 6 marinedrugs-19-00696-t006:** The cytotoxic activities of glycosides **1**–**4**, chitonoidoside E, cisplatin and chitonoidoside A (positive controls) against human erythrocytes, HeLa, DLD-1, THP-1 human cell lines.

Glycosides	ED_50_, µM, Erythrocytes	Cytotoxicity, ED_50_ µM			
		HeLa	DLD-1	THP-1	HL-60
Chitonoidoside E	0.45 ± 0.01	5.73 ± 0.10	8.93 ± 1.28	0.58 ± 0.07	5.73 ± 0.37
Chitonoidoside E_1_ (**1**)	0.64 ± 0.01	18.00 ± 0.59	34.12 ± 2.12	0.58 ± 0.04	9.97 ± 0.94
Chitonoidoside F (**2**)	1.17 ± 0.07	37.99 ± 1.36	71.11 ± 1.22	4.87 ± 0.47	41.23 ± 1.11
Chitonoidoside G (**3**)	0.65 ± 0.01	14.52 ± 1.08	12.44 ± 1.07	4.81 ± 0.46	8.23 ± 0.33
Chitonoidoside H (**4**)	0.89 ± 0.05	17.02 ± 1.18	36.62 ± 1.51	2.88 ± 0.23	8.13 ± 0.45
Chitonoidoside A	1.27 ± 0.03	39.48 ± 1.15	32.68 ± 2.56	2.93 ± 0.17	8.95 ± 0.35
Cisplatin	-	16.94 ± 0.25	> 80.00	56.12 ± 3.91	8.58 ± 0.54

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
