# Peer review of "Triterpene Glycosides from the Far Eastern Sea Cucumber Psolus chitonoides: Chemical Structures and Cytotoxicities of Chitonoidosides E1, F, G, and H"

_marinedrugs, 2021, doi:10.3390/md19120696_

Round 1

Reviewer 1 Report

This manuscript reports four new triterpene disulfated glycosides, chitonoidosides E1 (1), F (2), G (3), and H (4) from the sea cucumber to make input to the body of known chemical diversity and biological activity of huge collection of natural products. This paper is worth publishing after minor revision. I left comments on the pdf.

Author Response

We have carried out the followed changes outside of referees’ comments:

1) Lines 346–364, Section 3.2.: was rewritten, according to the plagiarism check request.

2) Lines 381, 383: 4.4.3 was replaced with4.3 and 4.4.6 was replaced with III.4.6.

3) Lines 437–451, Section 3.4. and lines 453–462, Section 3.5. were rewritten, according to the plagiarism check request.

4) Line 509: the word “Higher” was added.

5) Lines 278, 289, 293, 294: the data concerning the activity of the glycosides against DLD-1 and HL-60 cells were added to the text and to Table 6: “-1, monocytes THP-1 and leukemia promyeloblast HL-60 cells”; “adenocarcinoma HeLa and HL-60 cells, but“; “erythrocytes, HeLa and HL-60 cells”; “of DLD-1 and THP-1 cells”.

6) Lines 295–298: The sentence was added: The most significant difference in the activity of 2 and the other compounds of the series was observed for DLD-1 and HL-60 cell lines, confirming the diversity in the sensitivities of cancer cell lines.

 The answers to Reviewer 1 comments:

Note: “Cytotoxic activities against erythrocytes”" is confusing. Please use hemolytic activity instead, through out the manuscript.

Reply: It is fixed: Lines 24, 25: “Cytotoxic” was replaced with “Hemolytic”, the phrase “cytotoxic action against” was added.

Note: this part is summary of abstract. I suggest to explain "chitonoidosides" briefly citing previous papers.

Reply: Lines 45–55: This part of the manuscript was broadened and itemized according to the referee comment: “The Far Eastern sea cucumber Psolus chitonoides is the fourth chemically studied representative of the genus Psolus. The animals of this species contain complicated multicomponent mixture of triterpene glycosides. So, their separation and purification are difficult and time-consuming. Recently we have published a paper concerning isolation, structure elucidation and biologic activity of the series of the glycosides, named chitonoidosides A–E isolated from P. chitonoides [15]. These compounds have some interesting structural features such as new non-holostane aglycone lacking a lactone and having an 18(20)-epoxy cycle, 3-O-methylxylose residue in the carbohydrate chains of three of them, the sulfation of 3-O-methylxylose by C-4, and finally rather rare architecture of tetrasaccharide carbohydrate chain branched by C-4 Xyl1. As continuation of our research on the glycosides from this species four…..”

Note: Is not this "reversed phase?", Is this right information? I could not find it on internet [Polychrom, Biolar, Latvia].

Reply: Line 66: The sorbent Polychrom-1, used for a long time at the initial stages of glycosides purification, is powdered Teflon. So, it is initially hydrophobic (it was not reversed), therefore this variant of chromatography is called hydrophobic. There is a difference because all the reversed phases contain remains of polar groups because the full reverse is impossible.

Note: Probably "remove" is correct.

Reply: It is fixed. Line 67: “to delete the salts” is replaced with “to eliminate salts”.

Note: Please add reference of holotoxinogenin here.

Reply: It is fixed. Line 81: the reference [16] to holotoxinogenin was added, the further numbering of the references was changed.

Note: If you define M as this, M should include Na to explain m/z.

Reply: Line 69 in the .pdf file attached by the Referee: The molecular formulae of the natural glycosides contain Na+ as counter ion of sulfate groups that is pointed of in all the formulae in the text. It was established earlier by the determination of elemental composition of the sulfated glycosides that they exist in nature in the form of sodium salts. The m/z values of the ions [M2Na-Na]- are explained by the presence of two sulfate groups, one of which contains Na+ counter ion and another one is decationized. The m/z values of the two-charged ions [M2Na–2Na]2− corresponded to two decationized sulfate groups. Thus, the molecular formulae of the glycosides in the manuscript are correct.

Note: Please briefly explain the position was determined in previous paper.

Reply: Lines 90–100: the elucidation of carbohydrate chain structure of chitonoidoside E1 (1) was added: “and demonstrated six characteristic doublets of anomeric protons at δH 4.67–5.11 (J = 7.1–8.0 Hz) and six signals of anomeric carbons at δC 102.3–105.2. The analysis of the 1H,1H-COSY, 1D TOCSY, HSQC and ROESY spectra of 1 resulted in the assignment of the signals of two xylose residues, one quinovose, one glucose and 3-O-methylglucose as well as 3-O-methylxylose residues. The positions of sulfate groups were determined based on the deshielded due to α-shifting effect of sulfate group doubled signal at δC 67.1 that is characteristic for sulfated by C-6 glucopyranose units (C-6 signals of non-sulfated glucopyranose residues are usually observed at ~δC 61.2). The signals of C-5 MeGlc4 and C-5 Glc5 were shielded due to β-effect of sulfate groups to δC 75.3 and 75.5, correspondingly. These data indicate…..”

Note: Please describe key correlations in this sentence.

Reply: Lines 103–106: the key correlations, indicating the positions of glycosidic linkages were added: ”H-1 Xyl1/H-3 (C-3) of the aglycone, H-1 Qui2/H-2 (C-2) Xyl1, H-1 Xyl3/H-4 (C-4) Qui2, H-1 MeGlc4/H-3 (C-3) Xyl3, H-1 Glc5/H-4 (C-4) Xyl1 and H-1 MeXyl6/H-3 (C-3) Glc5 in the ROESY and HMBC spectra of 1, correspondingly”

Note: Bold and Italic are not clear on this manuscript. I recommend single under bar and double under bar, or adding asterisks instead.

Reply: Lines 125, 126 – footnotes under Table 1: In all our papers we traditionally use “Bold” highlighting for the glycosidic bond positions and “Italic” highlighting for the sulfate group positions. So, we would like to preserve the consistency in the styling of the papers. Moreover, the signal of C-6 Glc5 at δC 67.1 was not highlighted by “Italic” in the Table 1, the inaccuracy is fixed.

Note: These values make the table difficult to read. I suggest to remove these values as long as authors mention opinion about these values. Number of replicates are more important. Please add it instead.

Reply: Table 6: the fixed inaccuracies are highlighted by yellow. The values of standard errors (± 0.XX) are necessary requirements for the statistically processed data and also usually required by other referees. The number of experiments (triplicates) is given in the Sections 3.4, 3.5.

Reviewer 2 Report

The researcher from Vladivostok continue their study on triterpenoid glycosides from the Far Eastern sea cucumber Psolus chitonoides. Now they have isolated and identified four new sulfated glycosides, chitonoidosides E1, F, G, and H. Three of them have the same aglycone containing the 18,20-lactone moiety, holotoxinogenin, widely distributed in the sea cucumbers. The aglycone of the fourth compound has a less common 18,20-ether bond instead of the lactone. Two of the isolated compounds are branched hexaosides, other compounds have shortened oligosaccharide chains by one or two monosaccharides. The authors presented a tentative biosynthetic pathway of carbohydrate chains of chitonoisides of different groups. They concluded that glycosylation and sulfation are parallel and competitive processes. The structure elucidation of new compounds was professionally done and I have no doubt that the reported structures are correct.

I recommend the paper for publication in Marine Drugs essentially as is. I only suggest to replace in Abstract ‘disulfated glycosides’ with ‘sulfated glycosides’ since the glycosides E1 and G are not disulfated.

Author Response

The answers to the referee 2

Note: The researchers from Vladivostok continue their study on triterpenoid glycosides from the Far Eastern sea cucumber Psolus chitonoides. Now they have isolated and identified four new sulfated glycosides, chitonoidosides E1, F, G, and H. Three of them have the same aglycone containing the 18,20-lactone moiety, holotoxinogenin, widely distributed in the sea cucumbers. The aglycone of the fourth compound has a less common 18,20-ether bond instead of the lactone. Two of the isolated compounds are branched hexaosides, other compounds have shortened oligosaccharide chains by one or two monosaccharides. The authors presented a tentative biosynthetic pathway of carbohydrate chains of chitonoisides of different groups. They concluded that glycosylation and sulfation are parallel and competitive processes. The structure elucidation of new compounds was professionally done and I have no doubt that the reported structures are correct.

I recommend the paper for publication in Marine Drugs essentially as is. I only suggest to replace in Abstract ‘disulfated glycosides’ with ‘sulfated glycosides’ since the glycosides E1 and G are not disulfated.

Reply: The authors are very appreciative for so high estimation of our work. Concerning the recommendation: because chitonoidosides E1 (1) and G (3) are really disulfated compounds containing sulfate groups at C-6 MeGlc4 and C-6 Glc5, the correction is not needed.

Reviewer 3 Report

Although the manuscript has an interesting subject, in my opinion, needs to be revised before it can be accepted for publication.

Below please find my comments.

Introduction section - the Authors should try to make an effort to emphasize the importance of their studies. More details about the species used in the study are needed.

Results and Discussion is very important part of each manuscript published. In presented manuscript this section, especially bioactivity part, comprises too general explanations. Authors should discuss their results with more other scientific papers.

The conclusions should be integrated with more detailed results summarizing all the study and must reflect the innovation of this study and the perspectives.

Author Response

The replies on the referee 3 report

Note: Introduction section - the Authors should try to make an effort to emphasize the importance of their studies. More details about the species used in the study are needed.

Reply: Introduction section was broadened with the emphasis of the relevance of the studies of the glycosides: “Triterpene glycosides - characteristic secondary metabolites of the sea cucumbers, those studies provide significant input to the exploration of chemical diversity, properties and biological activity of huge collection of natural products which are the valuable and promising resource of new drugs and medicines [1–8]. The interest to these compounds also driven by their taxonomic specificity [9-11] as well as the possibility to reconstruct the sequences of biosynthetic transformations of the aglycones and carbohydrate chains during biosynthesis [12–13] and to define the peculiarities of «structure-activity relationships» based on the knowledge about their structural diversity [14]. All this indicate the relevance of the searching for new glycosides. The Far Eastern sea cucumber Psolus chitonoides is the fourth chemically studied representative of the genus Psolus.”

Note: Results and Discussion is very important part of each manuscript published. In presented manuscript this section, especially bioactivity part, comprises too general explanations. Authors should discuss their results with more other scientific papers.

Reply: Bioactivity part: The additional references discussing the cytotoxic and anticancer activity of the glycosides isolated from diverse species of the sea cucumbers and therefore having different structures as well as the comparison of the activity of this series of chitonoidosides with the previous one, are added to the part of the manuscript concerning biologic activity (section 2.2.):

Lines 265–271: “Cytotoxic activity of sea cucumber glycosides against different cell types and cell lines, including HeLa and THP-1 as frequently applied models, has been extensively studied [1, 19]. The deepen investigation on the cytotoxic compounds resulted in the clearing of the mechanisms of anticancer action of the glycosides [7, 8, 20]. Different tumor cell lines exhibit a differential sensitivity to the cytotoxic effects of sea cucumber glycosides depending on their chemical structures that can be of special interests for the therapy of certain types of cancer [21].”

Lines 279–282: The phrases “Human erythrocytes were more sensitive to the membranolytic action of tested compounds (Table 6) compared to the cancer cells similarly to the data concerning mouse erythrocytes [12, 13, 15, 17]. Monocytes cell line THP-1 and erythrocytes were comparably sensitive to the action of the glycosides.” were added.

Lines 292–294: The phrase “Latter two peculiarities are in good accordance with biologic activity of previous series of chitonoidosides – glycosides of P. chitonoides [15].” was added.

Note: The conclusions should be integrated with more detailed results summarizing all the study and must reflect the innovation of this study and the perspectives.

Reply: The conclusions were corrected and specified in accordance with the Reviewer comment:

Lines 486–494: The phrases “Noticeably that in the first series of the studied glycosides from P. chitonoides [15] 3-O-methylxylose residue sulfated by C-4 was found. This indicates the presence of specific sulfatase in this species of the sea cucumbers capable to attach sulfate group to C-4 of monosaccharides in pyranose form. The “structure-activity relationships” observed were following: tetraosides with shortened bottom semi-chain showed the weakest membranolytic effect, the activity of the glycosides having new type aglycone with 18(20)-ether bond instead of 18(20)-lactone was comparable with that for the substances having holostane-type aglycones. Hexaosides and tetraosides with linear carbohydrate chains (having bottom semi-chain) were the most active in the series of the glycosides from P. chitonoides.” were added.

Lines 497–500: The sentence “All the discussed data broaden the knowledge about structural diversity of triterpene glycosides from the sea cucumbers and move us to the understanding of the complicated process biosynthesis of this class of metabolites, which is so far a huge gap in knowledge for scientists.” was added.

Reviewer 4 Report

Authors isolated, characterised and evaluated four novel interesting glycosides from sea cucumber Psolus chitonoides. Structure of novel compounds is precisely documented. Basic cytotoxicity screening was performed and compounds shown membranolytic effect.

Article is very similar to the previously published one by the same authors (Marine Drugs 2021, 19, 449). Unfortunately, it is concepted rather as an appendix to this article than a independent article. Introduction is very short and does not provide sufficient insight into discussed issue. Conclusion is very short as well and does not reflect results of cytotoxicity assays.

For publication, following must be improved:

Add for introduction a brief overwiew of known Psolus glycosides and their biological activities. Structurally similar glycosides from other species should be briefly mentioned as well. 

All compounds shown strong membranolytic activity. This activity should be compared with other Psolus glycosides activity and glycosides membranolytic activity in general. This shoul be added to Dicussion part of the article. 

After fixing this flaws, the article will be worthy for publication.

Author Response

The replies on the referee 4 report

Authors isolated, characterised and evaluated four novel interesting glycosides from sea cucumber Psolus chitonoides. Structure of novel compounds is precisely documented. Basic cytotoxicity screening was performed and compounds shown membranolytic effect. Article is very similar to the previously published one by the same authors (Marine Drugs 2021, 19, 449). Unfortunately, it is concepted rather as an appendix to this article than an independent article.

Note: Introduction is very short and does not provide sufficient insight into discussed issue. Conclusion is very short as well and does not reflect results of cytotoxicity assays.

Reply: The section “Introduction” was broadened by additional references and their brief analysis and the section “Conclusions” was deepened and specified with the sentences above-mentioned in the answers to Reviewer’s comments.

Note: Add for introduction a brief overview of known Psolus glycosides and their biological activities. Structurally similar glycosides from other species should be briefly mentioned as well.

Reply: A brief overview of known Psolus chitonoides glycosides were added to the Introduction (Lines 51–56). Some hundreds triterpene glycosides are known from the sea cucumbers by now. These all are the structurally related compounds, so their mentioning in the text is impossible. Moreover, some reviews referenced in the text give the insights to the structural diversity of the glycosides. We have added the reference on latest review on this concern in the discussion of biosynthesis [23, Kalinin, V.I.; Silchenko, A.S.; Avilov, S.A.; Stonik, V.A. Progress in the studies of triterpene glycosides from sea cucumbers (Holothuroidea, Echinodermata) Between 2017 and 2021. Nat. Prod. Commun. 2021, 16, http://doi: 10.1177/1934578X211053934]

Note: All compounds shown strong membranolytic activity. This activity should be compared with other Psolus glycosides activity and glycosides membranolytic activity in general. This should be added to Discussion part of the article.

Reply: The comparison of the cytotoxic activity of all chitonoidosides isolated so far was done; the comparison of the activities of chitonoidosides and the other glycosides seems to be senseless, because of the using of different cell lines and different methodologies of the activity estimation. Moreover, the validated comparative analysis should be made in the frame on the same experiment. However, the tendencies of SAR can be revealed when compared the activities of the compounds in different experiments, but this issue of special research we only started recently: [Zelepuga, E.A.; Silchenko, A.S.; Avilov, S.A.; Kalinin, V.I. Structure-activity relationships of holothuroid’s triterpene glycosides and some in silico insights obtained by molecular dynamics study on the mechanisms of their membranolytic action. Mar. Drugs 2021, 19, 604. https://doi.org/10.3390/md19110604].

Round 2

Reviewer 4 Report

Authors modified manuscript due to recommendations and extended Introduction and Conclusion adding useful references. Therefore I recommend article for publication in present form.